# Determination of Waste Management Workers’ Physical and Psychological Load: A Cross-Sectional Study Using Biometric Data

**DOI:** 10.3390/ijerph192315964

**Published:** 2022-11-30

**Authors:** Itsuki Kageyama, Nobuki Hashiguchi, Jianfei Cao, Makoto Niwa, Yeongjoo Lim, Masanori Tsutsumi, Jiakan Yu, Shintaro Sengoku, Soichiro Okamoto, Seiji Hashimoto, Kota Kodama

**Affiliations:** 1Graduate School of Technology Management, Ritsumeikan University, 2-150 Iwakuracho, Osaka 567-8570, Japan; 2Merge System Co., Fukuoka 810-0041, Japan; 3School of Environment and Society, Tokyo Institute of Technology, Tokyo 108-0023, Japan; 4College of Science and Engineering, Ritsumeikan University, 1-1-1 Noji-Higashi, Shiga 525-8577, Japan; 5Center for Research and Education on Drug Discovery, The Graduate School of Pharmaceutical Sciences, Hokkaido University, Sapporo 060-0812, Japan

**Keywords:** waste management, psychological load, physical workload, occupational risks, biometric information

## Abstract

Waste management workers experience high stress and physical strain in their work environment, but very little empirical evidence supports effective health management practices for waste management workers. Hence, this study investigated the effects of worker characteristics and biometric indices on workers’ physical and psychological loads during waste-handling operations. A biometric measurement system was installed in an industrial waste management facility in Japan to understand the actual working conditions of 29 workers in the facility. It comprised sensing wear for data collection and biometric sensors to measure heart rate (HR) and physical activity (PA) based on electrocardiogram signals. Multiple regression analysis was performed to evaluate significant relationships between the parameters. Although stress level is indicated by the ratio of low frequency (LF) to high frequency (HF) or high LF power in HR, the results showed that compared with workers who did not handle waste, those who did had lower PA and body surface temperature, higher stress, and lower HR variability parameters associated with higher psychological load. There were no significant differences in HR, heart rate interval (RRI), and workload. The psychological load of workers dealing directly with waste was high, regardless of their PA, whereas others had a low psychological load even with high PA. These findings suggest the need to promote sustainable work relationships and a quantitative understanding of harsh working conditions to improve work quality and reduce health hazards.

## 1. Introduction

Waste management workers responsible for collecting, transporting, and sorting waste face various challenges [1,2]. They face numerous occupational risks, including long working hours, exposure to physical, chemical, mechanical, biological, ergonomic, and social risks, and frequent occupational accidents, which result in physical or mental illness [3]. The United States Department of Labor Occupational Safety and Health Administration includes toxic metals, crushing hazards, and dangerous energy release, among the potential hazards of working with waste [4]. In 2020, the frequency (number of fatalities and injuries due to occupational accidents per one million hours of total actual work) and intensity rates (number of lost workdays per one million hours of total actual work, i.e., the severity of the accident) of occupational accidents in Japan’s waste management industry were very high; approximately four times higher than the average of all other industries [5]. Accidents such as ‘falling and crashing’, ‘getting caught in or between’, and ‘tumbling’ are common. These represent the following: accidents due to falls from buildings and collapses and injuries resulting from a person being squeezed, caught, crushed, pinched, or compressed between two or more objects, or between parts of an object. Moreover, the waste work environment is rarely air-conditioned, and physical illnesses related to high body and environmental temperatures, such as heatstroke, are increasing in the field due to climate change.

Waste management facilities are mainly involved in processes such as crushing, sorting, compacting, and incineration. In areas where automated operations are difficult, waste work can be labour-intensive and dangerous. The Japan Construction Occupational Safety and Health Association (JCOSHA) regulates safety management efforts in the manufacturing and construction industries [6]. In labour-intensive industries such as construction, efforts are made to improve the sophistication of safety management using information and communication technologies for process refinements, such as abnormality detection and maintenance related to processes and equipment [7]. They monitor workers’ bio-information and behaviour, analyse risk, and evaluate the practicality of education and training. The use of information and communication technology in safety management is expected to improve the waste management working environment, which enhances the working conditions, quality of life, and health of workers, thereby ultimately reducing the industry’s impact on public health by contributing to environmental sustainability [1].

Significant advancements in wearable technology in recent years have enhanced opportunities to monitor the biometric data and physical condition of workers in the workplace. The use of wearable devices to observe working conditions through the measurement of heart rate (HR), body temperature, and physical activity (PA) is rapidly gaining popularity [8,9]. Assessing daily HR changes can help determine workload [10], mental state [11,12], and heart health [13]. Measuring heart rate variability (HRV) over a relatively short period (up to five minutes) can provide a highly accurate analysis [14]. Fatigue and stress resulting from physical and psychological strain can reduce satisfaction, well-being, and work efficiency in any workplace. An imbalance between physical and psychological resources and demands can threaten safety at work. Workers who continuously handle hazardous and potentially hazardous waste throughout the working day face daily risk, often under harsh conditions. The working conditions of high-risk workers have been monitored by observing their HR and HR interval variability. Tiwari et al. [15] and Hwang and Lee [16] analysed changes in HR during work to quantify workers’ physical and mental load. Changes in PA and biometric data are useful in quantitatively analysing the condition of subjects. For example, Jebelli et al. [17] examined the possibility of using physiological information collected from wearable devices to determine the physical and psychological state of workers in the construction industry. These workers also face a harsh working environment, and the characteristics of their work and working conditions pose high potential risks to their physical and mental health, including physical fatigue and mental stress. Considering that physical fatigue and induced mental stress can have detrimental effects on motivation, job satisfaction, productivity, quality, and safety, it is crucial to consider physical and mental health. Although it is crucial for waste management facilities to manage working conditions to maintain a healthy employment relationship with waste workers, there is little empirical evidence to support effective health management practices. Therefore, this study measured the physical and psychological load associated with the work of waste workers and identified factors that increase it. These results will contribute to the efficient management of workers in the waste industry.

## 2. Materials and Methods

### 2.1. Study Design

This cross-sectional study of workers in the waste treatment industry compared biometric information and workload index of workers completing two different types of tasks—industrial waste handling vs. non-industrial waste handling—at the same waste treatment facility. In this study, two groups were set up: “waste handling workers” and a “control group” who were not involved in waste handling tasks.

### 2.2. Hypotheses

Based on the research objectives and literature review, the following hypotheses were proposed:

**Hypothesis 1** **(H1):**
*Differences in the biological parameters of the workers are apparent in the comparison between waste treatment and non-waste treatment workers.*


**Hypothesis 2** **(H2):**
*Worker workload in a waste management facility is positively correlated with PA and body surface temperature.*


**Hypothesis 3** **(H3):**
*Using the measured parameters of waste workers, their workload and psychological load can be estimated.*


### 2.3. Ethical Approval

This study was approved by the Research Ethics Review Committee of Ritsumeikan University (BKC-2019-009) in accordance with the Declaration of Helsinki, United Nations Educational, Scientific and Cultural Organization’s Universal Declaration on the Human Genome and Human Rights.

### 2.4. Measurement Tools

The specifications of the measurement devices and equipment used in this study are listed in Appendix A, and the status of the measurement devices attached to the workers is shown in Appendix B. The waste management workers in this study wore uniforms to prevent direct contact with waste. The sensing wear used for data collection was placed under their uniform, next to their skin. It comprised a vest fitted with a biometric sensor to measure the HR and PA based on electrocardiogram (ECG) signals. The sensing wear was made of stretchable fabric, and the stretchable ECG electrodes were integrated with hardware for HR measurement. For the measurement of biometric data, we used the WHS-2 for HR measurement, a triaxial accelerometer (Union Tool Co., Ltd., Tokyo, Japan) to measure PA, COCOMI (Toyobo Co., Ltd., Osaka, Japan) as sensing wear, and the CC2650 data acquisition device (Texas Instruments, Dallas, TX, USA). WHS-2 was chosen because it can measure three-axis acceleration information and HR with high resolution. By measuring three-axis acceleration information, the workers’ body movements become clear. COCOMI was chosen because it can collect biometric information with high precision and is soft to the touch, with little foreign body sensation when it touches the skin. As in the case of previous studies, we deciphered the physical and mental state of workers using biometric information. The wet-bulb globe temperature (WBGT) of the working environment was measured using an AD-5696 data recorder (A&D Co. Ltd., Tokyo, Japan). The complete measurement system is presented in Figure 1. Using a low-energy Bluetooth device, the HR and triaxial acceleration data were first transmitted to the data acquisition device worn by the worker. Subsequently, the data were transmitted to a cloud server on the network using wireless access points (WiFi and 4G-based transfer devices) installed in the work area.

A flowchart of the background, objectives, and analysis in this study is presented in Figure 2, and the study was conducted according to this flowchart.

### 2.5. Participants

The data were collected from 3 to 4 September 2019, and 24 to 26 August 2020, at the Kyoto Environmental Conservation Corporation, a waste management facility in Fushimi-Ku, Kyoto, Japan. The participants were recruited from among the facility’s employees; workers who responded to the call from the facility took part in this experiment. The participants were defined as healthy adults aged 20–63; those with neurological or cardiovascular diseases were excluded. All the participants were men. A total of 29 workers (average age: 35, standard deviation (SD): 15.7 years; average body mass index (BMI): 21.5, SD 3.2; average work experience (EXP): 7.7, SD: 4.2 years) were included in the final sample.

Based on their job profile or responsibilities, the participants were allocated to two groups. Group A (*n* = 22) was responsible for the transport, dismantling, and sorting of industrial waste, and Group B (*n* = 7), for the maintenance and repair of the facility. The list of participants in each group is presented in Appendix C. Group A workers were responsible for unloading waste from trucks at the facility, manually arranging it, and then moving it by cart to the loading dock for incineration. In the waste sorting process, relatively small waste articles (not suitable for heavy machinery) were sorted manually, thus posing a continuous exposure risk. Group B workers were tasked with inspecting and repairing equipment at the facility and did not come into direct contact with waste. The difference in the number of Group A and Group B workers is due to the staffing of the facility where this experiment was conducted.

### 2.6. Protocols

Before data collection, the research team tested the sensing wear. They confirmed that it was a non-invasive measurement tool that posed no risk to the wearer, such as interference or discomfort with daily work tasks. An informed consent form was distributed to all the participants prior to data collection. It included an explanation of their rights and assured them of the confidentiality of their data. To minimise the risk of personal information leaks, personal identification codes (identifiers) were assigned to each participant during the experiment and data analysis.

### 2.7. Analysis Tools and Statistical Tests

Statistical analysis was conducted using SPSS Version 26 for Windows (IBM Corp., Armonk, NY, USA) and Excel add-in software Bell Curve (Social Survey Research Information Co., Ltd., Tokyo, Japan) for Excel version 3.21 (Microsoft Corporation, Redmond, WA, USA). In addition, the following statistical tests were performed on the three hypotheticals:

**Hypothesis 4** **(H4):**
*When comparing waste treatment workers to non-treatment workers, we tested whether there were significant differences in the biological parameters of the two populations of workers. The normality of data was tested using the Shapiro–Wilk and Kolmogorov–Smirnov tests (Appendix G). First, the normality between the data in the two populations was tested. If there was normality, a T-test was performed; if not, a Mann–Whitney U test was performed.*


**Hypothesis 5** **(H5):**
*To analyse the relationship between changes in biological parameters and temperature fluctuations that cause heat stroke and other physical ailments experienced by waste treatment workers, the effect of workers’ biological parameters on body surface temperature was analysed. Multiple regression analysis was performed, with the independent variable being the biological parameters of the workers and the dependent variable being body surface (BS) TEMP.*


**Hypothesis 6** **(H6):**
*To investigate whether the measured biological parameters of waste treatment workers could be used to estimate workload and psychological load, a multiple regression analysis was conducted with the workers’ biological parameters as independent variables and physical and psychological load as dependent variables. The multiple regression analysis conducted in this study investigated the presence or absence of multicollinearity among candidate variables and proceeded by combining dependent and independent variables with no multicollinearity.*


### 2.8. Body Temperature and HR

Human performance is affected by various environmental factors in a working system, including heat stress [18,19]. The measurement of body temperature is an effective way to observe heat stress in workers in hot environments. However, invasive methods such as rectal and oesophageal measurements are often impractical [20,21]. Humans need to dissipate excess heat generated by the body to maintain thermal equilibrium with their internal temperature maintained at approximately 37 °C. However, the uniforms provided by waste management companies often do not provide adequate protection, especially in the summer when workers are thinly clad. Thus, metabolic heat from the body, environmental factors (temperature, humidity, radiant heat, etc.), and the overall heat load from clothing can result in heat stress [22]. When the core body temperature increases due to poor heat dissipation, the physiological burden of heat can cause health problems such as heatstroke, heat exhaustion, and heat cramps [23]. The physiological process for maintaining body temperature becomes inadequate when personal risk factors are added to excessive environmental or metabolic heat stress, thus resulting in an elevated body temperature and pulse rate and weight loss due to dehydration [24]. If early signs of heat-related illness are disregarded, heat stress can lead to poor work performance and also heat-related injuries and death [25].

Studies using non-invasive wearable devices have found a significant correlation between human HR and body surface temperature measurements and the thermal environment [26,27,28,29]. Eggenberger et al. [30] investigated different exercises and clothing conditions in a hot and humid environment and found two measured parameters, HR and body surface temperature at the scapular region, suitable for predicting rectal temperature. The air layer trapped inside a garment creates a specific microclimate around the body [31], influencing perceived comfort [32]. The microclimate inside a garment influences human satisfaction and performance during activities [33]. Thus, HR and ambient temperature in clothing may be used as indicators of human thermal sensation and heat stress. However, few studies have investigated the effects of HR and ambient temperature on the heat stress of workers.

### 2.9. HRV Metrics

Heart rate variability (HRV) is not only related to its effect on physical load, but also to autonomic control, such as self-regulation and psychological and physiological stress, making it useful for psychological load analysis [14,34]. A low HRV index indicates inadequate coordination between the sympathetic and parasympathetic nervous systems and is a reliable predictor of future cardiovascular diseases [35,36]. Therefore, HRV analysis provides important information for the assessment of physical function and helps identify the risk of physical fatigue and debilitation [37]. The time-domain index of HRV quantifies the extent of variation in the heart rate interval (RRI), which is the time between successive heartbeats, and the frequency-domain index can be obtained from the power spectrum density of a specific frequency band of RRI data. HRV time-domain indicators include the standard deviation of the normal heart rate interval time (NN), called SDNN, the root mean square successive difference (RMSSD), the index of difference between adjacent normal heart rate intervals greater than 50 ms (NN50), and the index of the percentage of the difference between adjacent normal heart rate intervals greater than 50 ms (pNN50). The main frequency indicators are low frequency (LF) power and high frequency (HF) power [14].

In a previous study, RMSSD was related to workers’ perception of mental stress [38], with lower values indicating higher stress. The RMSSD metric is not significantly affected by the number of missing data points, thus indicating its robustness for assessing patients with poor data quality. The standard deviation of the RRI (SDRR) is calculated from the SD of the normal RR interval; a lower SDRR indicates lower HRV [39]. Taelman et al. [40], who explored the interaction between HRV and mental stress, reported significantly lower NN50 and mean RR intervals in mentally demanding tasks.

HRV metrics have shown promise in multiple applications for healthcare professionals, and several researchers have focused on this field [41,42,43,44]. Despite concerns about the validity of some HRV indices for measuring sympathetic and parasympathetic nerve activity [45,46], the significance of HRV analysis is supported by many studies, such as LF (0.04–0.15 Hz) power, HF (0.15–0.40 Hz) power, and their ratios. LF is considered an indicator of both sympathetic and parasympathetic nervous system activity, and its ratio to HF, an indicator of parasympathetic nervous system activity, can quantitatively indicate whether the sympathetic or parasympathetic nervous system is dominant. Such analysis has been used clinically to understand the tension state of sympathetic and parasympathetic nervous systems [47,48]. HRV is an objective measure of stress in healthcare workers, and Joseph et al. [49] found self-reported stress to be associated with proportionately elevated physiological levels. Their results provided convincing evidence that physicians who routinely perform surgical tasks assess their own stress under severe time constraints. The dissemination of objective and ecologically valid measures of stress may offer important clues for understanding stressful situations and reducing the psychological load [50,51].

HRV measurements are derived from RR data and are influenced by the length of the time series (number of data points), body posture, and activity type. In the present study, these factors were derived using the five-minute RRI, which provides values for each activity. Fuentes-García et al. [52] reported lower average HRV values even during relatively short periods of mental stress.

Outliers or error beats due to artefacts are typically found in the time series of heartbeat intervals and have no physiological significance. Artefacts can significantly distort measurements in the time and frequency domains and increase the power over a wide band of frequencies [53]. In HRV data, if the time series is clean and sufficiently long to calculate the power of a given frequency band, the value is valid for evaluating the power of LF and HF. For example, at least 2.5 min of clean data are required to evaluate LF power [54]. For a stable interpretation of autonomic function, Chen et al. [55] established that time-domain HRV indices (e.g., SDNN, RMSSD, pNN50) require one minute of short-term recordings, while the specifications for frequency-domain HRV indices (LF and HF) require at least three minutes of recorded data to be accurately measured. The HRV data analysed in this study comprised one segment of five minutes, which is considered sufficiently long.

The data set collected in this study consisted of evaluation data collected at five-minute intervals and was filtered based on previous studies to exclude the effect of artefacts on HR [56,57]. Briefly, at the centre point of a moving window of length *l*, data points outside the interval were excluded, and the mean of the data points within the moving window was calculated, excluding the centre point; *a* is a positive number equal to or less than one. In this study, *l* = 41 and *a* = 0.2 were used.

### 2.10. Workload (Percentage HR Reserve) and PA

To determine the physical load during waste management work, the HR and PA of workers were used as indicators. The basic relationship equations are presented in Appendix E and Appendix F. Percentage heart rate reserve (%HRR) is a measure of the physical load or pressure intensity associated with muscle activity [58]. Norton et al. [59] stated that 40–60% HRR lasting 30–60 min is equivalent to moderate physical load. Hwang and Lee [16] and Hashiguchi et al. [10], who focused on construction workers, noted that an HRR of 30–40% in all-day work continues to pose a health risk for workers. Equation (1) shows how it can be estimated:%HRR = (HR_working_ − HR_resting_)/(HR_maximum_ − HR_resting_) × 100 (%)(1)
where HR_working_ is the average working heart rate, HR_resting_ is the resting heart rate, and HR_maximum_ is the maximum heart rate based on age [60]. In this study, HR_resting_ was defined as the lowest stable five-minute heart rate in the work break.

A small accelerometer (WHS-2) attached to the sensing wear placed on the chest was used to obtain the composite acceleration along three axes (vertical axis: *X*, horizontal axis: *Y*, and vertical axis: *Z*) [23]. Five-minute average values were used to calculate the amount of PA and to observe the overall intensity of workers’ movements during working hours. The intensity of PA was calculated from the five-minute average values, which is a predictor of health status [61]. Detailed movement data could be obtained from the biometric data parameters, thus providing an understanding of the movement intensity of workers concerning work actions performed during the working day (e.g., lifting a load, carrying a load and walking, standing, and squatting) [62].

## 3. Results

### 3.1. Data Collection

We collected data on the participants’ HR, physical acceleration, and body surface temperature at various times during the working day. The WBGT of each group was measured in their respective work areas. A total of 1945 data points, measured approximately every five minutes, were collected from the 29 participants. The monitoring of workers was conducted throughout the working day, which included a one-hour break, and observations were made from 9:00 a.m. to 4:30 p.m. The data are presented in Appendix D. We collected self-reported information on age, years of work experience, height, and weight from the workers.

The research team did not monitor participants’ behaviour but recorded their work activities using two cameras installed near each work area. Prior to the experiment, we told the participants that we would not measure their operating skills and instructed them not to deviate from their daily working routines, thereby avoiding the Hawthorne effect [63].

### 3.2. Descriptive Statistics and Intergroup Comparisons

The normality of data was tested using the Shapiro–Wilk and Kolmogorov–Smirnov tests (Appendix G). In both tests, the null hypothesis assumes that the data set is normally distributed at *p* = 0.05 [64]. However, there was a slight discrepancy between the tests; all the data except HR satisfied the condition of normal distribution.

The Mann–Whitney U test does not require a normally distributed data set [65], and its null hypothesis is ‘no difference between the two groups at a significance level of 0.05.’ At *p* < 0.05, the null hypothesis is rejected, indicating a statistically significant difference in the distribution of data for each group [39]. Statistically significant differences were found between workers for PA, BS TEMP, WBGT, NN50, pNN50, and RMSSD of HRV time-domain parameters and LF power, HF power, and LF/HF of HRV frequency domain parameters. For all the other parameters, no statistically significant differences were found between workers. The results of the analysis are presented in Table 1.

In addition, the time-domain parameters related to the autonomic nervous system, NN50, pNN50, and RMSSD, were low in waste management workers, thus indicating a decrease in parasympathetic activity or a state of relaxation. In the frequency-domain parameters, they may also have lower LF power associated with the parasympathetic nervous system and higher LF/HF, indicating stress. The ratio of LF power to HF power (LF/HF) is a classic indicator of sympathetic balance [66]. This value represents the overall balance between the sympathetic and parasympathetic nervous systems. A high value indicates sympathetic dominance, while a low value indicates parasympathetic dominance.

### 3.3. Relationship of Workers’ Characteristics, Body Load, and HRV Index with Body Surface Temperature

The data collected in each work environment were merged and analysed for their effects on the body surface temperature of the workers. Multiple regression analysis was performed to evaluate significant relationships between each parameter. The results are presented in Table 2. In the multiple regression analysis, we checked for multicollinearity among the independent variables (worker characteristics, body load, and HRV index). Age, EXP, and BMI for worker characteristics, HR and PA for workload, SDNN, CVRR, and RMSSD for HRV time domain, WBGT for the work environment, and LF power, HF power, and LF/HF for HRV frequency domain all had variance inflation factors (VIFs) of less than 10. Age, HR, PA, and WBGT showed a significant relationship with BS TEMP.

As shown in Table 3a,b, higher HR and PA were positively correlated with BS TEMP. Table 3c,d reveals that WBGT and AGE had a positive effect on BS TEMP, but the coefficient of determination was low, which suggests a small but statistically significant effect.

### 3.4. Indexes of Physical and Psychological Load

The data collected from each work environment were merged to evaluate the relationship among worker characteristics, PA, BS TEMP, HRV index, and physical and psychological loads as dependent variables. Table 4 presents the results. Prior to the multiple regression analysis of workload, multicollinearity was assessed with VIF, and no multicollinearity among the independent variables was confirmed. Among the independent variables of workload %HRR, BMI, PA, BS TEMP, Age, EXP, and LF power had a VIF of one to two, suggesting no possibility of multicollinearity. Multiple regression analysis of psychological load LF/HF indicated that among the independent variables of psychological load LH/HF, the indices RRI, PA, BS TEMP, Age, experience, and LF power had low VIF, thus suggesting no possibility of multicollinearity. RRI in workload %HRR and BMI in psychological load LH/HF were excluded from the independent variables in each multiple regression equation because VIF > 10. For each workload %HRR and psychological load LF/HF ratio, statistically significant independent variables were adopted in the multiple regression equations.

### 3.5. Relationship to Workload %HRR and Psychological Load LF/HF

First, we obtained a multiple regression equation that demonstrated the effects of the independent variables PA, BS TEMP, and age on the workload. The regression equation, as shown in Equation (2), has an adjusted R^2^ of 0.449. The results of the analysis are shown in Table 5, where the changes in PA, BS TEMP, and LF power all had a positive impact on the workload of the workers. The standard regression deviations of the independent variables revealed approximately the same values, and the percentage of influence on the workload was equal.
Workload: %HRR = 37.1 × PA + 2.34 × BS TEMP + 0.269 × AGE − 70.1(2)

Subsequently, we obtained a multiple regression equation showing the effects of the independent variables RRI, BS TEMP, LF power, and EXP on psychological load. The multiple regression equation, as shown in Equation (3), has an adjusted R^2^ of 0.356. The results of the analysis are given in Table 6, wherein changes in BS TEMP and LF power had a positive impact on the psychological load of workers, whereas RRI and EXP had a negative impact. The standard regression bias of the independent variables suggested that the greatest effect was due to the LF power related to the autonomic nervous system.
Psychological load: LF/HF = −0.0065 × RRI − 0.186 × BS TEMP + 0.0003 × LF power − 0.205 × EXP + 0.356(3)

The biometric data of waste treatment workers were analysed to obtain a multiple regression equation for estimating their load at work, and the results established that BS TEMP was significantly affected by both workload (%HRR) and psychological load (LH/HF).

## 4. Discussion

### 4.1. Principal Findings

This study investigated the effects of workers’ individual characteristics and biometric indices on their physical and psychological loads during waste-handling operations. The results indicated significant differences in the workers’ HRV time-domain parameters (NN50, pNN50, and RMSSD), and HRV frequency parameters (LF power, HF power, and LF/HF) when comparing waste handling and non-waste handling operations. Compared with workers who did not handle waste, those who did had a lower PA and BS TEMP; lower HRV parameters (NN50, pNN50, RMSSD) associated with higher psychological load, and higher LF/HF and LF power were associated with stress. There were no significant differences in HR, RRI, and %HRR. These results suggest that waste disposal workers work under a high psychological load, regardless of their HR and workload %HRR due to their work.

Workers’ workloads and psychological loads were estimated using measurement parameters. Although studies have analysed the relationship between body temperature, WBGT, and HR [67,68], this study confirmed that body surface temperature could be used for this estimation. Workload %HRR was estimated from PA, body surface temperature, and workers’ age (adjusted R^2^ = 0.449), and psychological load LF/HF was estimated from the autonomy-influenced RRI, LF power, BS TEMP, and years of experience (adjusted R^2^ = 0.356). PA and BS TEMP had a positive effect on workload %HRR [17], and the three independent variables, including workers’ age, affected workload almost equally. RRI had a negative effect, and LF power and BS TEMP had a positive effect on the psychological load of LF/HF of workers [69,70]. There was a tendency to report a lower psychological load with increasing years of work experience. These findings support all three hypotheses. Although RMSSD, NN50, and pNN50 showed significant differences when comparing workers who handled waste to those who did not, they were not parameters that determined psychological load LF/HF. Orsila et al. [38] confirmed the relationship between RMSSD and mental stress during relatively long periods of time in the morning, afternoon, and evening among employees of an electronics company. However, the environment and conditions of work were different compared with this study, which focused on waste work, and there were differences in the results of each of these studies.

### 4.2. Physical and Psychological Load in Waste Management Workplaces

UEMAE et al. [71] observed the relationship of temperature and humidity near the skin to changes in HR, a physiological index, and stated that changes in skin temperature affect HR and that the factors are related to HR’s autonomous thermoregulatory function [71]. The sympathetic activity of the autonomic nervous system can be used to estimate the intensity of physical and psychological loads, such as stress [72]. Therefore, it can be inferred that the body surface temperature affects %HRR and HRV, which are load indicators related to the work environment. This inference is based on the relationships shown in Equations (2) and (3). We also found that waste workers are exposed to various levels of stress due to the handling of waste and the temperature of their work environment. Our results indicate that uncertainty about workers’ psychological load and its acceptable limits can be resolved by analysing the environment in which waste is handled and the temperature conditions of the workers.

In the estimation of %HRR, body surface temperature was affected as an explanatory variable, but there was no effect of environmental temperature on WBGT. This relationship indicates a low correlation between body surface temperature and WBGT, as evident from Table 3c. Studies have reported the effects of WBGT on biological information [18,73]. However, it can be inferred that WBGT near each work site may not be related to workers’ body temperature and that analyses using body surface temperature can show changes in worker HR more accurately. Furthermore, the LF/HF estimation revealed that workers handling waste had higher sympathetic nervous activity (i.e., higher LH/HF and LF power) than parasympathetic nervous activity (i.e., lower RRI, SDRR, RMSSD, and HF power) related to the catatonic state. In addition, workers’ HRV indices were low, which aligned with previous findings [74,75], suggesting that increased work-related stress leads to decreased parasympathetic activity and increased sympathetic activity. Thus, the LF/HF ratio of workers provides insight into their work stress levels.

The participants of this study were exposed to low physical load and high mental stress. This indicates that waste workers may experience higher psychological stress due to the characteristics of their jobs, such as taking extra care in regards to the safety of their environment and working with hazardous waste, than those engaged in hard physical labour. Eggenberger et al. [30] demonstrated that the frequency of exposure of waste handlers to hazardous materials significantly increased perceptual stress, and workers with a higher frequency of exposure reported 2.14 times more stress than those with a lower exposure. As waste management companies have high psychological job demands, it is necessary to assess workers’ job-related psychological factors. It would be useful for waste management companies to use this research tool to regularly and quantitatively assess the physical and psychological load of workers. Instead of using questionnaires and interviews to assess workers’ health and well-being, this research tool will enable companies to more effectively assess workers’ health and provide an appropriate working environment.

### 4.3. Approaches to Worker Stress Reduction and Worker Management

In the estimation of workers’ psychological load, experience was inversely correlated with LF/HF (standard β of EXP = −0.018). Experience is negatively correlated with LF/HF because, as workers gain more experience, they have more knowledge about dealing with the waste. Thus, the psychological load may be reduced by improving inexperienced workers’ knowledge of health literacy, waste to be handled, protective gear, and tools and education and training. Studies using questionnaire surveys of waste workers have established that worker demographics (e.g., education level, marital status, and the number of children) and working conditions (e.g., shift work, work hours, and income) are significantly related to psychological load levels [76,77,78]. Waste workers have a high turnover rate, which makes it difficult to collect personal information in some cases. Our results confirm that work stress can be estimated independent of worker demographics and working conditions, and may be used for worker management independent of country or industry.

### 4.4. Theoretical and Practical Contributions

This study makes the following theoretical contributions. The psychological load LH/HF by HRV and %HRR as workload were used to quantify the workload of waste management workers. In addition, equations for estimating these loads using physical and psychological loads from HR and biometric data, including PA and body surface temperature, were developed.

Regarding practical contributions, the study found that the physical and psychological loads of waste workers, which have not been clarified previously, differ depending on the type of work. This evidence and monitoring of workers will assist the waste management industry to reduce health risks and enhance the working conditions of workers, and it can suggest new management concepts to companies by considering these factors from the perspective of health psychology.

### 4.5. Limitations

This study has several limitations. First, physical characteristics and age were not equally balanced in the sample. Although physical characteristics affecting workload and stress vary with age and gender [79,80], the results may not be generalisable to all waste workers as the study was conducted on healthy men aged 20–63 years. The waste management industry in Japan is dominated by men, and future studies should include women in their sample. Second, this study was a cross-sectional analysis, with data collected during the working days for seven days. Additionally, the mental and physical status of workers at the beginning of the workday was not measured. Since nutritional status, family environment, past stress conditions, lack of sleep, and lack of physical activity (sports) affect personnel activities, observing workers’ conditions over a longer period of time could yield more useful results. Investigation of background factors and initial conditions should be the subject of future research. Finally, as frequency-based metrics have been reported to represent the balance between sympathetic and parasympathetic activity more accurately [81], it is vital to improve the quality of HR interval recordings in wearable devices. Certain data collected in this study had missing HR data intervals, which affected the selection of HRV metrics and necessitated the removal of a few participants from the statistical analysis. Future studies aimed at further improving sensing wear and wearable technology and enhancing recording quality (e.g., further minimising motion artefacts) are essential to investigate large samples over time. Such improvements will increase the usefulness of these devices.

## 5. Conclusions

In this study, physical and psychological loads in the workplace were investigated among workers at a waste management facility in Japan. The workers performed their daily tasks in an environment exposing them to high stress depending on environmental temperature and task. Those who directly handled waste had lower PA, lower body surface temperature, and lower HRV parameter values associated with the autonomic nervous system than maintenance workers at the facility. Workload and stress among waste management workers, which has not been investigated in previous studies, suggested a significant relationship between HR variability, body surface temperature, PA, age, and years of experience. The workers’ body surface temperature was significantly related to HR, PA, environmental temperature, and age. The psychological load LF/HF was affected by the HRV parameters, RRI and LF power, body surface temperature, and years of experience.

Regular monitoring of biometric and physical information such as HR, body surface temperature, and activity can help understand the working environment of workers with high physical and psychological loads and promote the maintenance of sustainable relationships among them in harsh manufacturing and construction environments [10]. A quantitative understanding of working conditions in harsh work environments is important for improving work quality and reducing health hazards. Waste management companies can reduce the negative impacts on the environment and public health by promoting sustainable working relationships.

## Figures and Tables

**Figure 1 ijerph-19-15964-f001:**
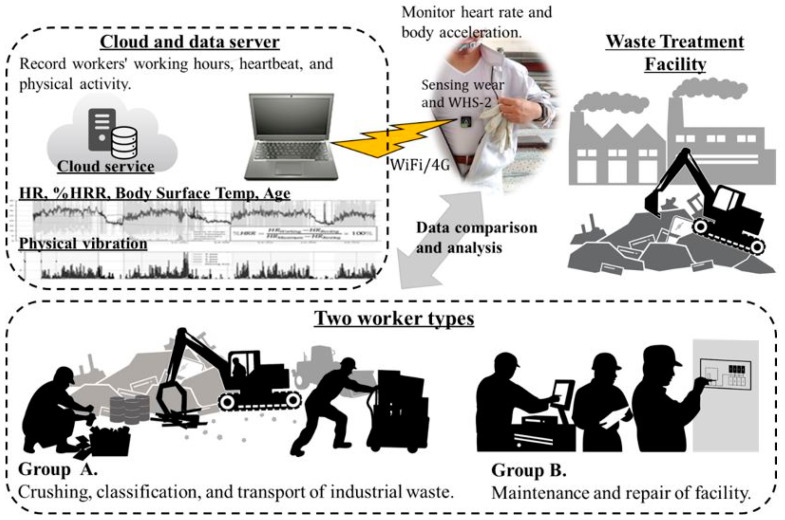
Overall configuration of the measurement system. HR: Heart rate; %HRR: Heart rate reserve.

**Figure 2 ijerph-19-15964-f002:**
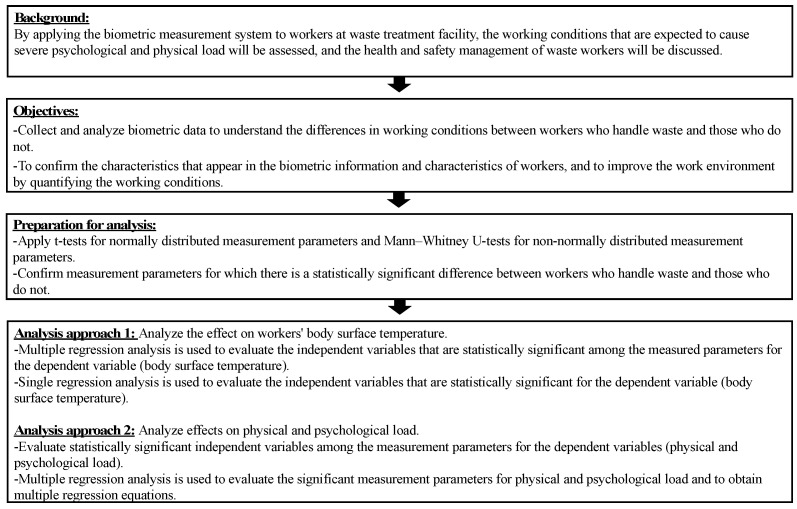
Study background, objectives, and flowchart of analysis.

**Table 1 ijerph-19-15964-t001:** Means, standard deviations, and *p*-values of each parameter for both work groups.

Parameters (Unit)	Workers in Waste Management Facility		*p*-Value between Groups
	1. Group A	2. Group B	
Workers’ characteristics			
Age (years)	35.5 ± 16.8	33.3 ± 12.7	0.38
EXP (years)	7.23 ± 4.6	9.29 ± 1.9	0.07
BMI (%)	21.2 ± 3.5	22.6 ± 1.4	0.23
Physical and environmental			
HR (bpm)	94.2 ± 13.2	94.3 ± 16.2	0.21
%HRR (%)	21.6 ± 12.7	23.2 ± 14.5	0.15
PA (mG)	222.4 ± 116.3	261.8 ± 126.2	<0.001
BS TEMP (°C)	31.4 ± 1.86	31.9 ± 2.02	<0.001
WBGT (°C)	30.0 ± 0.72	31.2 ± 0.57	<0.001
HRV time-domain			
RRI (ms)	650.3 ± 94.5	653.8 ± 106.6	0.21
SDRR (ms)	15.4 ± 16.2	25.6 ± 16.2	0.13
CVRR	0.039 ± 0.003	0.039 ± 0.039	0.14
NN50	20.2 ± 20.8	30.5 ± 30.2	<0.001
pNN50 (%)	0.058 ± 0.058	0.083 ± 0.087	<0.001
RMSSD (ms)	21.7 ± 8.00	24.3 ± 9.87	<0.001
HRV frequency-domain			
LF power (ms^2^)	1070.5 ± 8853	695.6 ± 1120	<0.001
HF power (ms^2^)	263.0 ± 537.6	740.6 ± 695.6	<0.001
LF/HF	3.74 ± 7.58	2.38 ± 2.65	<0.001

Note: Group A = waste treatment workers; Group B = non-waste treatment workers; EXP = experience; BMI = body mass index; HR = heart rate; %HRR = percent heart rate; PA = physical activity; BS TEMP = body surface temperature; WBGT = wet bulb globe temperature; HRV = heart rate variability; RRI = heart beat interval; SDRR = standard deviation of heart beat interval; CVRR = coefficient of variation of heart beat interval; NN50 = index of difference between adjacent normal heart rate intervals greater than 50 ms; pNN50 = index of the percentage of the difference between adjacent normal heart rate intervals greater than 50 ms; RMSSD = root mean square successive difference; HRV = heart rate variability; LF = low frequency; HF = high frequency. Among the parameters, only HR showed a normal distribution, so HR was *t*-tested. Other parameters were tested for significance using the Mann–Whitney U test.

**Table 2 ijerph-19-15964-t002:** Relationship between independent (worker characteristics, workload, HRV index) and dependent (BS TEMP) variables.

Independent Variable	Dependent Variable: BS TEMP			
	Β	S.E.	Std. β	*p*-Value
AGE	0.011	0.0037	0.0962	0.02
EXP	−0.011	0.012	−0.0249	0.36
BMI	−0.0853	0.0649	−0.0572	0.19
HR	0.060	0.0085	0.412	<0.0001
PA	1.78	0.398	0.111	<0.0001
WBGT	0.335	0.062	0.120	<0.0001
SDNN	−0.0016	0.0382	−0.0017	0.97
CVRR	12.8	24.6	0.0212	0.60
RMSSD	−0.0067	0.0072	−0.030	0.35
LF power	0.00001	0.00001	0.0096	0.69
HF power	0.00001	0.00001	−0.0210	0.34
LF/HF	0.012	0.0065	−0.0341	0.17

Note: BS TEMP = body surface temperature; EXP = experience; BMI = body mass index; HR = heart rate; PA = physical activity; WBGT = wet bulb globe temperature; SDRR = standard deviation of normal heart rate interval; CVRR = coefficient of variation of heart beat interval; RMSSD = root mean square successive difference; HRV = heart rate variability; LF = low frequency; HF = high frequency; B = slope (coefficient) of regression equation.

**Table 3 ijerph-19-15964-t003:** Effect of independent variables on body surface temperature.

**(a) Effect of Independent Variable (HR) on BS TEMP**				
**Independent Variable**	**Dependent Variable: BS TEMP**			
	**Estimated**	**S.E.**	***t*-Value**	***p*-Value**
HR	0.0686	0.0029	23.7	<0.001
(Intercept)	30.0	0.0738	406.8	<0.001
Adjusted R^2^	0.223			
F static value	559.9			<0.001
**(b) Effect of Independent Variable (PA) on BS TEMP**				
**Independent Variable**	**Dependent Variable: BS TEMP**			
	**Estimated**	**S.E.**	***t*-Value**	***p*-Value**
PA	4.96	0.342	14.5	<0.001
(Intercept)	30.4	0.088	344.6	<0.001
Adjusted R^2^	0.198			
F static value	210.8			<0.001
**(c) Effect of Independent Variable (WBGT) on BS TEMP**				
**Independent Variable**	**Dependent Variable: BS TEMP**			
	**Estimated**	**S.E.**	***t*-Value**	***p*-Value**
WBGT	0.596	0.066	9.10	<0.001
(Intercept)	13.5	1.98	6.83	<0.001
Multiple R^2^	0.048			
F static value	82.9			<0.001
**(d) Effect of Independent Variable (AGE) on BS TEMP**				
**Independent Variable**	**Dependent Variable: BS TEMP**			
	**Estimated**	**S.E.**	***t*-Value**	***p*-Value**
AGE	0.0094	0.0026	3.61	<0.001
(Intercept)	31.8	0.104	298.63	<0.001
Multiple R^2^	0.0062			
F static value	813.1			<0.001

Note: BS TEMP = body surface temperature; HR = heart rate; PA = physical activity; WBGT = wet bulb globe temperature.

**Table 4 ijerph-19-15964-t004:** Relationships between dependent and independent variables representing physical and psychological load.

Independent Variables	Dependent Variables					
	%HRR			LF/HF		
	B	S.E.	*p*-Value	B	S.E.	*p*-Value
BMI	−0.099	0.145	0.50			
RRI				−0.0089	0.0014	<0.001
PA	12.5	1.26	<0.001	−2.25	1.07	0.04
BS TEMP	0.772	0.080	<0.001	−0.163	0.0646	0.01
AGE	0.270	0.0082	<0.001	0.0046	0.0085	0.59
EXP	0.008	0.045	0.86	−0.218	0.0324	<0.001
LF power	0.00001	0.0001	0.61	0.0003	0.00001	<0.001

Note: %HRR = percent heart rate; LF = low frequency; HF = high frequency; BMI = body mass index; RRI = heart beat interval; PA = physical activity; BS TEMP = body surface temperature; EXP = experience; B = slope (coefficient) of regression equation.

**Table 5 ijerph-19-15964-t005:** Relationship between workers’ biological information and %HRR.

Independent Variable	Dependent Variable: %HRR					VIF
	Partial β	S.E.	Standard β	*t*-Value	*p*-Value	
PA	37.1	1.94	0.340	19.2	<0.001	1.11
B.S. TEMP	2.34	0.122	0.339	19.1	<0.001	1.11
AGE	0.269	0.0134	0.339	20.0	<0.001	1.01
(Intercept)	−70.1	3.75	−	−18.7	<0.001	−
Adjusted R^2^	0.449					
F static value	529.0					
Sig.	<0.001					

**Table 6 ijerph-19-15964-t006:** Relationship between workers’ biological information and LF/HF.

Independent Variable	Dependent Variable: LF/HF	VIF
Partial β	S.E.	Standard β	*t*-Value	*p*-Value
RRI	−0.0065	0.0012	−0.147	−6.11	<0.001	1.27
B.S. TEMP	0.186	0.0549	0.088	2.74	0.006	1.26
LF power	0.0003	0.00001	0.561	18.3	<0.001	1.01
EXP	−0.205	0.0425	−0.018	−8.11	<0.001	1.01
(Intercept)	12.9	2.22	−	5.81	<0.001	−
Adjusted R^2^	0.356					
F static value	156.6					
Sig.	<0.001

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
