# Peer review of "Determination of Waste Management Workers’ Physical and Psychological Load: A Cross-Sectional Study Using Biometric Data"

_ijerph, 2022, doi:10.3390/ijerph192315964_

Round 1
Reviewer 1 Report
This paper is well written and organized. I have a few suggestions that can help improve the clarity further given the large number of variables analyzed in this study and several minor comments at the end.
Overall, I think it would be helpful to include a summary table of all variables with columns including variable abbreviation, full name, and description. In that way, readers can easily access the information in an easy-to-read format, and the authors won’t need the long note under the current Tables (e.g., Table 1-4).
Another good visualization to include is a roadmap of the analysis as well as the logic/objective behind such sequence of analysis. Because there are a number of correlation analysis and regressions, using different sets of variables, I got lost when reading the manuscript.
Last, the significance of this manuscript would improve if the authors can provide some discussion about potential policies that could learn from this study and help workers’ physical and psychological health. Right now, the discussion is mostly on direct study results. It would be more interesting to extend more from there and talk about potential ways that this study could help improve the workers’ health in practice.
Minor comments:
Abstract – “The results showed … with stress” line 24-17 is not very clear. It’s also hard to understand “LF and FH and LF power” in that sentence without an appropriate context. I would simplify the sentence into just “psychological load” but before that in another sentence, I would describe how “psychological load” is measured using different variables. This should help present it more clearly.
Line 43, what is “:” and what is “the weight of disaster”?
Line 46, could you be more descriptive on the accidents? Right now, I am sure what these quotes mean.
Line 58, there’s a special character which should not belong here. Also, not sure how “effectiveness of …” fits here
Line 88, the term “non-industrial waste handling” is confusing because it can be understood as “handling non-industrial waste, but still handling waste” but this should not be what the authors intended to say. Instead, I would keep the use of the relative terms consistent throughout the manuscript. For example, sometimes the authors use “non-treatment workers”, sometimes “non-waste treatment workers”. Maybe define it clearly upfront with “waste handling workers” and the “control group”, which does not involve waste handling tasks. And then just refer to the two groups as “waste handling” vs “control”
Line 62 not sure how that can contribute to reducing the impact on the “environment” given the context of this study
Line 106 is “measurement device” repeated info?
Line 116 make it clear that the three-axis acceleration info is about body movement
Line 188 not sure what it means by “were pre-confirmed multicollinearity”. Could you provide more detail?
Section 2.9 first paragraph is hard to read. I would reorganize it by first defining HRV (and preferably in mathematical forms; a figure may also help) and then explain why it is used to measure physical and psychological stress.
Line 235 citations or reasons are needed to support the argument that RMSSD is not significantly affected by …
Line 267, suggest changing it to “at least 2.5 min of clean data are required for evaluating LF power.”
Line 279 provides an explanation why l = 41 is chosen and does it mean 41 5-min segments?
L294 HR_resting is defined twice. Define only once with accurate info.
Table 1 note says the parameters were tested using Mann-Whitney U test, but I remember earlier in the manuscript, it says when variable is normally distributed, then t-test is used. Could you clarify?
Table 2: what’s “B”? similarly in Table 4, make a note about beta to be clear.
Line 379 when you say “no multicollinearity”, do you mean “no perfect multicollinearity”?
Table 4 is not clear on what “*” means. The table caption is also confusing
Line 451 do the results support this argument that HR is more affected by BS TEMP than humidity? If not, a citation is needed.
Author Response
2022 年 11 月 19 日
レビュアー各位
まず、感謝の意を表したいと思います。あなたのコメントや提案は、私たちの論文を改善するために非常に貴重です.
以下に示すすべてのコメントと提案に従って、改善に最善を尽くしたと考えています。私たちの対応がお客様の期待と意図に沿うものであることを願っています。
どうぞよろしくお願いいたします。
敬具、
児玉 浩太 PhD
立命館大学大学院技術経営研究科
〒567-8570 大阪府茨城県岩倉町2-150
+81-72-665-2448
ID |
コメントと提案 |
応答 |
レビュアー 1-1 |
I think it would be helpful to include a summary table of all variables with columns including variable abbreviation, full name, and description. In that way, readers can easily access the information in an easy-to-read format, and the authors won’t need the long note under the current Tables (e.g., Table 1-4). |
Thank you for reading the details and your suggestions to improve this paper. We added a summary list of variables as an Appendix H. The notes under each table were retained because they are necessary for the tables to be individually readable.
|
Reviewer 1-2 |
Another good visualization to include is a roadmap of the analysis as well as the logic/objective behind such sequence of analysis. Because there are a number of correlation analysis and regressions, using different sets of variables, I got lost when reading the manuscript. |
Thank you for your comment. It's just as you said. We summarize in Figure 2 describing the study background, objectives, and flowchart of analysis in this study. |
Reviewer 1-3 |
The significance of this manuscript would improve if the authors can provide some discussion about potential policies that could learn from this study and help workers’ physical and psychological health. Right now, the discussion is mostly on direct study results. It would be more interesting to extend more from there and talk about potential ways that this study could help improve the workers’ health in practice. |
Thank you very much for your advice. We added the following description in 4. Discussion regarding the potential of this study to help improve the health of workers in practice.
… As waste management companies have high psychological job demands, it is necessary to assess workers’ job-related psychological factors. It would be useful for waste management companies to use this research tool to regularly and quantitatively assess the physical and psychological load of workers. Instead of using questionnaires and interviews to assess, companies may be able to improve workers’ health and well-being and provide an appropriate working environment for workers. |
Reviewer 1-4 |
Abstract – “The results showed … with stress” line 24-17 is not very clear. It’s also hard to understand “LF and FH and LF power” in that sentence without an appropriate context. I would simplify the sentence into just “psychological load” but before that in another sentence, I would describe how “psychological load” is measured using different variables. This should help present it more clearly. |
Thank you very much for your advice. It's just as you said. Based on the reviewer's advice, we have modified the description in the Abstract as follows.
… Although stress level is indicated by the ratio of low frequency (LF) to high frequency (HF) or high LF power in HR, the results showed that compared with workers who did not handle waste, those who did had lower PA and body surface temperature, higher stress, and lower HR variability parameters associated with higher psychological load. There were no significant differences in HR, … |
Reviewer 1-5 |
Line 43, what is “:” and what is “the weight of disaster”? |
Thank you for your comment. “:” means i.e. and “weight” means severity. We modified to the following sentence.
… (number of lost workdays per one million hours of total actual work:, i.e.; the weight severity of the disaster) …
|
Reviewer 1-6 |
Line 46, could you be more descriptive on the accidents? Right now, I am sure what these quotes mean. |
Thank you for your comment. It was a difficult sentence to understand as you said. Therefore, we added the following sentence for a more detailed description of the accident in Japan.
… Accidents such as ‘falling and crashing,’ ‘getting caught in or between,’ and ‘tumbling’ are common. These represent the following: accidents due to falls from buildings and collapses and Injuries resulting from a person being squeezed, caught, crushed, pinched, or compressed between two or more objects, or between parts of an object. Moreover, …
|
Reviewer 1-7 |
Line 58, there’s a special character which should not belong here. Also, not sure how “effectiveness of …” fits here |
Thank you for your comment. We have removed the typo. え
We modified to the following description.
… They are monitoring workers’ bio-information and behaviour, and analysing risk, effectiveness practicality of education and training. |
Reviewer 1-8 |
Line 88, the term “non-industrial waste handling” is confusing because it can be understood as “handling non-industrial waste, but still handling waste” but this should not be what the authors intended to say. Instead, I would keep the use of the relative terms consistent throughout the manuscript. For example, sometimes the authors use “non-treatment workers”, sometimes “non-waste treatment workers”. Maybe define it clearly upfront with “waste handling workers” and the “control group”, which does not involve waste handling tasks. And then just refer to the two groups as “waste handling” vs “control” |
Thank you for reading the details and your suggestions to improve this paper. We followed the reviewer's advice and defined "waste handling group" and "control group" in 2.1, Study Design for clarity.
… —industrial waste handling vs non-industrial waste handling—at the same waste treatment facility. In this study, two groups were setup: "waste handling workers" and "control group", who were not involved in waste handling tasks. |
Reviewer 1-9 |
Line 62 not sure how that can contribute to reducing the impact on the “environment” given the context of this study |
Thank you for your comment. It's just as you said. We modified the detailed description to the following lines.
…, thereby ultimately reducing the industry’s impact on the environment and public health on public health by contributing to environmental sustainability [1].
|
Reviewer 1-10 |
Line 106 is “measurement device” repeated info? |
Thank you for your comment. We modified a detailed description.
The specifications of the measurement device and infrastructure used in this study are listed in Appendix A, and the measurement device in Appendix B and the status of the measurement devices attached to the workers is shown in Appendix B.
|
Reviewer 1-11 |
Line 116 make it clear that the three-axis acceleration info is about body movement |
Thank you for your comment. We added the implication of the measurement by three-axis acceleration.
… By measuring three-axis acceleration information, the workers’ body movements become clear. …
|
Reviewer 1-12 |
Line 188 not sure what it means by “were pre-confirmed multicollinearity”. Could you provide more detail? |
Thank you for your comment. It was a difficult sentence to understand as you said. Therefore, on line 188, we added and modified the following detailed description.
The independent variables used were pre-confirmed multicollinearity. The multiple regression analysis conducted in this study investigated the presence or absence of multicollinearity among candidate variables and proceeded by combining dependent and independent variables with no multicollinearity.
|
Reviewer 1-13 |
Section 2.9 first paragraph is hard to read. I would reorganize it by first defining HRV (and preferably in mathematical forms; a figure may also help) and then explain why it is used to measure physical and psychological stress. |
Thank you for reading the details and your suggestions to improve this paper. It was a difficult sentence to understand as you said. Therefore, based on the reviewer's suggestion that the text in the first paragraph of Section 2.9 was difficult to read, we have added an explanation of the HRV.
HRV is the generic term for heart rate variability; HR variability is not only related to its effect on physical load, but also to autonomic control, making it useful for psychological load analysis. Specifically, indices measured by HRV are related to aspects of health by autonomic function, …
|
Reviewer 1-14 |
Line 235 citations or reasons are needed to support the argument that RMSSD is not significantly affected by … |
Thank you for your comment. You are right, there was a lack of information. Therefore, we added descriptions to 4. Discussion regarding RMSSD not affecting psychological load, using the citation of [38], following the reviewer's point.
…These findings support all three hypotheses. Although RMSSD, NN50, and pNN50 showed significant differences when comparing workers who handled waste to those who did not, they were not parameters that determined psychological load LF/HF. Orsila et al [38] confirmed the relationship between RMSSD and stress during relatively long periods of time in the morning, afternoon, and evening among employees of an electronics company. However, the environment and conditions of work were different compared to this study, which focused on waste work, and there were differences in the results of each of these studies.
|
Reviewer 1-15 |
Line 267, suggest changing it to “at least 2.5 min of clean data are required for evaluating LF power.” |
… For example, approximately at least 2.5 minutes of clean data must be are required for evaluated for LF power [54].
|
Reviewer 1-16 |
Line 279 provides an explanation why l = 41 is chosen and does it mean 41 5-min segments? |
Thank you for your comment. The text was difficult to understand as you indicated. Therefore, in line 279, we have modified the description to the following line.
…and the mean of the data points within the moving window was calculated, excluding the centre point; a is a positive number equal to or less than one. In this study, l = 41 and a = 0.2 were used. In this study, l=41 and a=0.2 were also used based on previous studies [56,57].
|
Reviewer 1-17 |
L294 HR_resting is defined twice. Define only once with accurate info. |
Thank you for your comment. In line 294, we have modified the description to the following line.
In this study, HRresting was defined as the lowest stable five-minute heart rate in the work break. |
Reviewer 1-18 |
Table 1 note says the parameters were tested using Mann-Whitney U test, but I remember earlier in the manuscript, it says when variable is normally distributed, then t-test is used. Could you clarify? |
Thank you very much for your advice. It's just as you said. In the note to Table 1, we have corrected the description to the following line.
The parameters were tested for significant differences using the Mann Whitney U test. Among the parameters, only HR showed a normal distribution, so HR was t-tested. Other parameters were tested for significance using the Mann-Whitney U test. |
Reviewer 1-19 |
Table 2: what’s “B”? similarly in Table 4, make a note about beta to be clear. |
Thank you for your comment. In Tables 2 and 4, B denotes the slope (coefficient) of the regression equation. We have added the annotations. |
Reviewer 1-20 |
Line 379 when you say “no multicollinearity”, do you mean “no perfect multicollinearity”? |
Thank you very much for your advice. It was a difficult sentence to understand as you said. Therefore, let me modify it to the following sentence.
Prior to the multiple regression analysis of workload, multicollinearity was assessed with VIF, no multicollinearity among the independent variables was confirmed. |
Reviewer 1-21 |
Table 4 is not clear on what “*” means. The table caption is also confusing |
Thank you for your comment. Due to a typo, the * will be deleted. For clarity, we would like to change the title of Table 4 to "Relationships between dependent and independent variables representing physical and psychological load. |
Reviewer 1-22 |
Line 451 do the results support this argument that HR is more affected by BS TEMP than humidity? If not, a citation is needed. |
Thank you very much for your advice. Please let us modify the description using citations, as it was not clear. In line 451, Although we did not measure humidity near the skin in our study, we think that skin temperature has a significant effect on HR, as pointed out in the cited reference.
UEMAE ら [71] は、皮膚付近の温度と湿度と、生理学的指標である HR の変化との関係を観察し、皮膚温度の変化が HR に影響を与え、その要因が HR の自律的な体温調節機能に関連していると述べました [71]。自律神経系の交感神経活動は、ストレスなどの身体的および心理的負荷の強度を推定するために使用できます [72]。…
|
Reviewer 2 Report
The work is well presented and is of high interest nowaday.
To stimulate the discussion, I would draw attention to the fact that the authors do not consider the "initial state" of the worker. I mean during the monitoring period, no assumption or measurement has been made on the initial (at the beginning of the working turnation) Psico-Physical condition of the worker. Nutrition, home facts, previous stress conditions, bad sleep, lack of physical activity (sport) can influence the speculations made on the cold measurement of HR activity. By the way, for sure the study presents a good approach to a theme that needs to be further investigated using ICT systems nowaday available.
Author Response
2022 年 11 月 19 日
レビュアー各位
まず、感謝の意を表したいと思います。あなたのコメントや提案は、私たちの論文を改善するために非常に貴重です.
以下に示すすべてのコメントと提案に従って、改善に最善を尽くしたと考えています。私たちの対応がお客様の期待と意図に沿うものであることを願っています。
どうぞよろしくお願いいたします。
敬具、
児玉 浩太 PhD
立命館大学大学院技術経営研究科
〒567-8570 大阪府茨城県岩倉町2-150
+81-72-665-2448
ID |
コメントと提案 |
応答 |
レビュアー 2-1 |
To stimulate the discussion, I would draw attention to the fact that the authors do not consider the "initial state" of the worker. I mean during the monitoring period, no assumption or measurement has been made on the initial (at the beginning of the working turnation) Psico-Physical condition of the worker. Nutrition, home facts, previous stress conditions, bad sleep, lack of physical activity (sport) can influence the speculations made on the cold measurement of HR activity.
|
Thank you for reading the details and your suggestions to improve this paper. Based on the reviewer's advice, we have added the following description to 4.5. Limitations.
… 第二に、この研究は横断的分析であり、データは 7 日間の営業日に収集されました。さらに、就業開始時の労働者の精神的および身体的状態は測定されませんでした。栄養状態、家庭環境、過去のストレス状態、睡眠不足、運動不足(スポーツ)などは人事活動に影響を与えるため、長期的に労働者の状態を観察することで、より有用な結果が得られる可能性があります。背景要因と初期条件の調査は、今後の研究の対象となるでしょう。 年間を通して長期間にわたってデータを収集すると、より決定的な結果が得られる場合があります。
|
Reviewer 3 Report
Dear Authors,
This paper is generally good, and I have only some minor corrections to proposed. The most important one the presentation of the results in tables 3,5 and 6 in graphical form, if possible, to better asses the relationship between them. I most also note that a significant fraction of the information is in the appendix that are not available to the reviewer. Therefore, this reduces the value of this review.
I have noted some additional minor corrections that are unrelate to the scientific value of the paper.
-line 58 : a hiragana character え is present on this line
-lines 181-126 : It appears this section should appears sooner in the text
-line 195 : 37°C an unbreakable space should separate the number and the symbol 37 °C
Author Response
November 19, 2022
Dear Reviewer,
First of all, we’d like to express our gratitude to you. Your comments and suggestions are invaluable for us to improve our paper.
We believe that we did our best in improvements according to all of your comments and suggestions as below, and we hope that our responses meet your expectations and your intentions.
We highly appreciate your cooperation.
Warm Regards,
Kota Kodama, PhD
Graduate School of Technology Management, Ritsumeikan University
2-150, Iwakura-cho, Ibaraki, Osaka, 567-8570, Japan
+81-72-665-2448
ID |
Comments and Suggestions |
Response |
Reviewer 3-1 |
…the presentation of the results in tables 3,5 and 6 in graphical form, if possible, to better assess the relationship between them. |
Thank you very much for your advice, but we are discussing this in a multiple regression analysis and consider the current table to be the best. In Tables 3, 5, and 6, please let us show them in table form for the sake of consistency in the multiple regression analysis.
|
Reviewer 3-2 |
I most also note that a significant fraction of the information is in the appendix that are not available to the reviewer. |
Our sincere apologies. We were delayed in submitting the appendix. Please confirm the additional Appendix again. |
Reviewer 3-3 |
-line 58 : a hiragana character え is present on this line. |
Thank you for your comment. We have removed the typo. え |
Reviewer 3-4 |
-lines 118-126 : It appears this section should appears sooner in the text |
Thank you for reading the details and your suggestions to improve this paper. Following the reviewer's advice, we moved on to the description of the parameters in the last paragraph of the Introduction.
… Tiwari et al. [15] and Hwang and Lee [16] have analysed changes in HR during work to quantify workers' physical and mental load. However, there are few reports on the work and the physical and psychological load of waste workers. Changes in PA and biometric data are useful in quantitatively analysing the condition of subjects. For example, Jebelli et al. [17] examined the possibility of using physiological information collected from wearable devices to determine the physical and psychological state of workers in the construction industry. They also face a harsh working environment, and the characteristics of their work and working conditions pose high potential risks to their physical and mental health, including physical fatigue and mental stress. Considering that physical fatigue and induced mental stress can have detrimental effects on motivation, job satisfaction, productivity, quality, and safety, it is crucial to consider physical and mental health. Although it is crucial for waste management facilities to manage working conditions to maintain a healthy employment relationship with waste workers, … |
Reviewer 3-5 |
-line 195 : 37°C an unbreakable space should separate the number and the symbol 37 °C |
Thank you for your comment. We modified the typo.
… such that their internal temperature is approximately 37 °C. However, … |
|
|
|